# Joint Association between Sedentary Time and Moderate-to-Vigorous Physical Activity with Obesity Risk in Adults from Latin America

**DOI:** 10.3390/ijerph20085562

**Published:** 2023-04-18

**Authors:** Eduardo Rossato de Victo, Mauro Fisberg, Dirceu Solé, Irina Kovalskys, Georgina Gómez, Attilio Rigotti, Lilia Yadira Cortes, Martha Cecilia Yépez-Garcia, Rossina Pareja, Marianella Herrera-Cuenca, Clemens Drenowatz, Diego Christofaro, Timoteo Araujo, Danilo Silva, Gerson Ferrari

**Affiliations:** 1Disciplina de Alergia, Imunologia Clínica e Reumatologia do Departamento de Pediatria da Universidade Federal de São Paulo, São Paulo 04023-062, Brazil; 2Centro de Excelência em Nutrição e Dificuldades Alimentares (CENDA), Instituto Pensi, Fundação José Luiz Egydio Setubal, Hospital Infantil Sabará, São Paulo 01228-200, Brazil; 3Departamento de Pediatria, Universidade Federal de São Paulo, São Paulo 04023-061, Brazil; 4Faculty of Medical Sciences, Pontificia Universidad Católica Argentina, Buenos Aires C1107, Argentina; 5Department of Biochemistry, School of Medicine, Universidad de Costa Rica, San José 11501-2060, Costa Rica; 6Department of Nutrition, Diabetes and Metabolism, School of Medicine, Pontificia Universidad Católica, Santiago 8330024, Chile; 7Department of Nutrition and Biochemistry, Pontificia Universidad Javeriana, Bogotá 110231, Colombia; 8Colegio de Ciencias de la Salud, Universidad San Francisco de Quito, Quito 17-1200-841, Ecuador; 9Instituto de Investigación Nutricional, Lima 15026, Peru; 10Centro de Estudios del Desarrollo, Universidad Central de Venezuela (CENDES-UCV)/Fundación Bengoa, Caracas 1053, Venezuela; 11Division of Sport, Physical Activity and Health, University of Education Upper Austria, 4020 Linz, Austria; 12Faculdade de Ciências e Tecnologia, Universidade Estadual Paulista, Presidente Prudente 01049-010, Brazil; 13Faculdades Metropolitanas Unidas, São Paulo 03021-000, Brazil; 14Department of Physical Education, Federal University of Sergipe, São Cristóvão 49100-000, Brazil; 15Facultad de Ciencias de la Salud, Universidad Autónoma de Chile, Providencia 7500912, Chile

**Keywords:** sitting time, sedentary behavior, obesity, physical activity, South America

## Abstract

Recent studies have shown various relationships between physical activity and the incidence of obesity, but this study critically explored the association of sedentary time (ST) and moderate-to-vigorous physical activity (MVPA) with obesity risk in adults from eight Latin American countries. ST and MVPA were assessed with accelerometers and stratified into 16 joint categories. Multivariate logistic regression models were used. The obesity risk indicators evaluated were body mass index (BMI), waist circumference (WC), and neck circumference (NC). Quartile 4 of ST and ≥300 min/week of MVPA was associated with lower odds of BMI compared to quartile 1 of ST and ≥300 min/week of MVPA. Quartile 1 of ST and 150–299 min/week of MVPA, quartile 1 of ST and 76–149 min/week MVPA, quartile 3 of ST and 76–149 min/week MVPA, and quartiles 1, 2, and 3 of ST with 0–74 min/week MVPA were associated with higher odds of high WC compared to quartile 1 of ST and ≥300 min/week of MVPA. Quartile 3 of ST and 150–299 min/week of MVPA, quartiles 1 and 3 of ST and 76–149 min/week of MVPA, and quartile 1 of ST and 0–74 min/week MVPA were associated with higher NC compared to quartile 1 of ST and ≥300 min/week of MVPA. This study suggests that achieving the MVPA recommendations will likely protect against obesity, regardless of ST.

## 1. Introduction

Epidemiologic studies show robust evidence of the benefits of physical activity on several health outcomes and the prevention of obesity [1,2,3]. The 2020 World Health Organization physical activity guidelines recommend that adults engage in at least 150 to 300 min/week of moderate physical activity, 75 to 150 min/week of vigorous physical activity, or an equivalent combination of both intensities [4]. Using objective methods to assess moderate-to-vigorous physical activity (MVPA) in Latin America—rare in this region where most previous research has relied on self-reported instruments [5,6]—a previous study showed that 41% of adults do not meet these physical activity guidelines, ranging from 27% in Chile to 48% in Costa Rica [7]. 

Sedentary behavior, on the other hand, refers to the time spent at intensities of less than 1.5 metabolic equivalents (METs) in a sitting, lying, or reclining position during waking hours [8]. Overall, 78% of adults from Latin America spent ≥8 h/day in sedentary activities [9,10]. Spending excessive time in sedentary pursuits has been associated with an increased risk for obesity, triglycerides, metabolic syndrome, cardiovascular disease, and mortality [11].

However, considering that both MVPA and sedentary time (ST) tend to coexist within human beings, understanding their joint associations and health-enhancing potential is important for the development of health recommendations. For instance, there are still gaps in the evidence to support ST guidelines for people with different levels of physical activity [12]. Even though high-income countries have already accumulated data on the joint association of MVPA and ST and have introduced recommendations in their public health guidelines [13], ST has not yet received the due attention of public authorities in middle-income countries (i.e., Latin America), which may be attributed to the scarcity of data.

Although the prevalence of overweight and obesity in Latin America has reached 60%, one of the highest in the world, constituting the region with the highest percentage of physically inactive people, few studies of representative samples have been conducted using accelerometers to measure ST and MVPA to study their combined association with obesity and other health indicators [14,15,16]. For instance, Gonze et al. showed that replacing 10 min blocks of ST with light and/or MVPA reduced obesity rates, especially when supplementing ST with MVPA in Brazilian adults [17]. In another study to examine the dose–response associations of physical activity and ST with body mass index (BMI), with data from Brazil, Colombia, and Mexico, the results showed a negative association between MVPA and BMI, especially in men [18].

Studies that investigated the joint association of physical activity and sedentary time are still limited in Latin America. There are few studies in this region that have measured physical activity with objective methods, such as the accelerometer. Most studies used self-report as a physical activity assessment tool. Different intensities of physical activity in combination with ST are explicitly recognized to have different graded associations with various health outcomes. According to the gap exposed by the Guidelines Development Group (a group of specialized scientists and public health professionals who inform the elaboration of the Guidance Plan on Physical Activity and Sedentary Behavior), studies are needed to investigate the joint association between physical activity and ST and its health outcomes [19]. Therefore, the aim of this study was to analyze the joint association of ST and MVPA with obesity indicators in adults from Latin American countries. The hypothesis of the study is that reaching the MVPA recommendations can significantly minimize the risks of obesity for those with lower ST; however, for those with excess ST, the risk would be slightly higher.

## 2. Materials and Methods

### 2.1. Study Design and Sample

The Latin American Study of Nutrition and Health (Estudio Latinoamericano de Nutrición & Salud-ELANS) is an observational, epidemiologic, multinational, and cross-sectional study conducted across eight countries within the Latin American region (Argentina, Brazil, Chile, Colombia, Costa Rica, Ecuador, Peru, and Venezuela), which represents 60% of the total number of countries in this region [20].

### 2.2. Data Collection Procedure

A multistage random sample of 9000 adolescents and adults was taken. The sample participants were between 15 and 65 years old, stratified by geographic location, gender, age, and socioeconomic level. To maintain a homogeneous population, only urban areas were included, which represent 80% to 90% of each country’s population. The sample size was calculated with a confidence level of 95% and a maximum error of 3.49%. More details on sampling procedures have been previously published [20].

Data collection occurred between September 2014 and February 2015. The overarching ELANS protocol was approved by the Western Institutional Review Board (#20140605) and is registered in Clinical Trials (#NCT02226627). Ethical approval was obtained from each local Institutional Review Board. All aspects of the study were in accordance with the Declaration of Helsinki. The ethical review boards also approved each protocol specific to the location of the participating institutions, and informed consent was obtained from the participants. Details about the ELANS study, participant sampling, recruitment strategies, and participants have been previously described [16,20].

A total of 9218 (52.2% women) participants (aged 15.0 to 65.0 years) were enrolled in the ELANS study. In this study, only participants who used accelerometers to measure ST and physical activity (2737 participants) were considered [21]. In addition, adolescents aged between 15 and 19 years were excluded due to the different cut-off points used for obesity risk in adolescents and adults. Therefore, the present study included a final sample of 2406 participants between 20 and 65 years of age.

### 2.3. Sedentary Time and Moderate-to-Vigorous Physical Activity

The ActiGraph GT3X+ accelerometer (Pensacola, FL, USA) was used to assess time spent (min/day) in sedentary pursuits and MVPA. The GT3X+ has been shown to provide valid and reliable estimates of ST and MVPA in adults in laboratory and in free-living conditions [22,23,24].

The accelerometer was placed on right mid-axillary line of the participants using an inelastic belt. Participants were asked to wear the device over a period of seven days, from the time they woke up in the morning until the time they went to bed at night, except when engaging in water-based activities. Participants were asked to complete an accelerometer log-sheet describing their wear-time per day. Days with ≥10 h of recorded wear time were considered valid [25]. The minimum days/times considered acceptable to be included in the analyses were five days with ≥10 h of wear time after the removal of sleep time, including at least one weekend day. After excluding the night-time sleep period, nonwear waking time was defined by any sequence of ≥60 consecutive minutes of 0 activity counts [26]. Details on the accelerometer data have been published elsewhere [7,20,21].

Participants received the accelerometers during inhouse visits, and the research team went to participants’ homes to retrieve the device. The team downloaded the data using the latest available version of the ActiLife software (version 6.0; ActiGraph, Pensacola, FL, USA). Data were collected at a sampling rate of 30 Hz and downloaded in epochs of 60 s [27]. ST time and MVPA were defined as <100 activity counts/min and ≥1952 counts/min, respectively [22,28].

Due to the inconsistency regarding cut-off values for ST in association with health outcomes [29], ST was categorized by quartiles (0–488.8 min/day: quartile 1; 488.9–558.1 min/day: quartile 2; 558.2–633.7 min/day: quartile 3; and >633.7: quartile 4). According to the physical activity recommendations for MVPA, the sample was divided into 4 groups: 0–74, 75–149, 150–299, and ≥300 min/week [30,31]. Finally, we jointly categorized participants in the following 16 categories of ST and MVPA: (1) quartile 1 of ST and 0–74 min/week of MVPA, (2) quartile 2 of ST and 0–74 min/week of MVPA, (3) quartile 3 of ST and 0–74 min/week of MVPA, (4) quartile 4 of ST and 0–74 min/week of MVPA, (5) quartile 1 of ST and 75–149 min/week of MVPA, (6) quartile 2 of ST and 75–149 min/week of MVPA, (7) quartile 3 of ST and 75–149 min/week of MVPA, (8) quartile 4 of ST and 75–149 min/week of MVPA, (9) quartile 1 of ST and 150–299 min/week of MVPA, (10) quartile 2 of ST and 150–299 min/week of MVPA, (11) quartile 3 of ST and 150–299 min/week of MVPA, (12) quartile 4 of ST and 150–299 min/week of MVPA, (13) quartile 1 of ST and ≥300 min/week of MVPA, (14) quartile 2 of ST and ≥300 min/week of MVPA, (15) quartile 3 of ST and ≥300 min/week of MVPA, and (16) quartile 4 of ST and ≥300 min/week of MVPA.

### 2.4. Obesity Risk

In each country, obesity risk (BMI and waist and neck circumferences) was measured with participants wearing light clothing and no shoes, using standard procedures [20].

Body height (to the nearest 0.5 cm) was measured using a Seca 213^®^ portable stadiometer (Hamburg, Germany) at the end of a deep inhalation with the participant’s head positioned in the Frankfort Plane. Body weight (to the nearest 0.1 kg) was measured using a portable electronic scale (Seca 213^®^, Hamburg, Germany), with an upper limit of 200 kg [32,33]. BMI was calculated in kg/m^2^, and participants were classified into underweight/eutrophic (≤24.9 kg/m^2^) or overweight/obese (≥25.0 kg/m^2^) [33].

Waist circumference was measured on the skin with an inelastic tape after removing accessories, such as belts and straps, located in the abdominal region. Participants were standing, with their feet together and arms positioned next to the body. Waist circumference was measured midpoint between the last rib and the iliac crest after taking a normal breath. According to the reference values proposed for the Latin American adult population, men and women with a waist circumference equal to or greater than 94 and 90 cm, respectively, were classified as having abdominal obesity [34].

Neck circumference (in centimeters) was measured using an inelastic measuring tape at the point just below the larynx (thyroid cartilage) and perpendicular to the long axis of the neck (with the ribbon line at the front of the neck at the same height as the ribbon line at the back of the neck) [35]. Men with a neck circumference >38.5 cm and women with a neck circumference >34.5 cm were classified as obese [36]. Two measurements for neck and waist circumferences were performed, and the average was used for analyses.

### 2.5. Sociodemographic Variables

Sociodemographic characteristics were assessed using standard questionnaires during face-to-face interviews. Responses were obtained regarding country, sex, age, marital status, socioeconomic status, and race/ethnicity. Although the socioeconomic classification differs across participating countries, the socioeconomic level was grouped into three broad classification levels (low, medium, and high) for all countries. Race/ethnicity was classified as mixed/Caucasian, black, white, and other. Details have been published elsewhere [20].

### 2.6. Statistical Analysis

Weighting was performed according to sociodemographic characteristics, sex, socioeconomic level, and country. Descriptive statistics included mean with standard deviation (SD) and percentages. We used multivariable logistic regression models (odds ratio: OR with the respective 95% confidence interval: 95%CI) to estimate the joint association of ST and MVPA (independent variable) with obesity risk factors (BMI and waist and neck circumferences) adjusted for country, sex, age, marital status, socioeconomic status, and race/ethnicity. Values of *p* < 0.05 were considered statistically significant. Data analyses were performed with IBM SPSS, v.26 (SPSS Inc., IBM Corp., Armonk, New York, NY, USA).

## 3. Results

There were no significant differences (*p* > 0.05) between the participants of the current study and the ELANS general sample by sex, age, marital status, socioeconomic status, and race/ethnicity. The sample consisted of 2406 participants (53.6% women), with a mean age of 39.3 years (SD: 12.9). Overall, 53.9% were married, 20.9% lived in Brazil, 50.7% were of a low socioeconomic status, and 51.1% reported mixed/Caucasian race/ethnicity. The mean ST and MVPA were 566.5 (SD: 120.5) and 34.02 (SD: 22.9) min/day, respectively. Overall, 65.5% of the participants were overweight or obese, and 44.1% and 42.8% were above the waist and neck circumference cut-points (Table 1). 

Table 1 presents the characteristics of the participants according to the two extreme categories of ST and MVPA (i.e., quartile 4 of ST and 0–74 min/week of MVPA vs quartile 1 of ST and ≥300 min/week of MVPA). Participants with quartile 4 of ST and 0–74 min/week of MVPA were more likely to be female, of low socioeconomic status, and classified as overweight/obese than participants with quartile 1 of ST and ≥300 min/week of MVPA (Table 1).

Figure 1 shows the proportion of participants in each of the 16 categories of ST and MVPA. The proportions of participants in the two extreme categories of ST and MVPA were 9.1% for quartile 1 of ST and ≥300 min/week of MVPA and 5.7% for quartile 4 of ST and 0–74 min/week of MVPA.

Pooled estimated associations of the 16 categories of ST and MVPA with obesity risk are shown in Figure 2, Figure 3 and Figure 4. Participants in quartile 4 of ST and ≥300 min/week of MVPA (OR: 0.54; 95%CI: 0.33, 0.89) showed lower odds for overweight/obesity compared to participants in quartile 1 of ST and ≥300 min/week of MVPA. No other significant associations were observed for overweight/obesity based on BMI (Figure 2).

According to the classification of obesity by waist circumference, the participants in quartile 1 of ST and 150–299 min/week of MVPA (OR: 1.75; 95%CI: 1.13, 2.71) showed higher odds for obesity than those in quartile 1 of ST and ≥300 min/week of MVPA. Participants in quartile 1 of ST and 76–149 min/week of MVPA (OR: 2.05; 95%CI: 1.27, 3.33) and those in quartile 3 of ST and 76–149 min/week of MVPA (OR: 1.61; 95%CI: 1.02, 2.54) also showed higher odds for obesity compared to participants in quartile 1 of ST and ≥300 min/week of MVPA. Participants in quartile 1 of ST and 0–74 min/week of MVPA (OR: 1.87; 95%CI: 1.01, 3.44) and those in quartile 2 of ST and 0–74 min/week of MVPA (1.74; 95%CI: 1.01, 2.99) and quartile 3 of ST and 0–74 min/week of MVPA (OR: 1.79; 95%CI: 1.05, 3.06) showed higher odds for obesity compared to participants in quartile 1 of ST and ≥300 min/week of MVPA (Figure 3). Participants in quartile 1 of ST and 150–299 min/week of MVPA (OR: 1.58; 95%CI: 1.05, 2.36) showed higher odds for obesity based on neck circumference than those in quartile 1 of ST and ≥300 min/week of MVPA. Participants in quartile 1 of ST and 76–149 min/week of MVPA (OR: 2.09; 95%CI: 1.30, 3.36) and those in quartile 3 of ST and 76–149 min/week of MVPA (OR: 1.58; 95%CI: 1.01, 2.48) had higher odds for obesity compared to participants in quartile 1 of ST and ≥300 min/week of MVPA. Those in quartile 1 of ST and 0–74 min/week of MVPA (OR: 1.88; 95%CI: 1.03, 3.42) also showed higher odds for obesity than participants in quartile 1 of ST and ≥300 min/week of MVPA (Figure 4).

## 4. Discussion

The present study aimed to analyze the joint association between ST and MVPA with different indicators of obesity (BMI and waist and neck circumferences) in adults from eight Latin American countries. The results show that, regardless of the time spent sedentary, MVPA tends to be negatively associated with obesity indicators.

In this study, adults who achieved 300 min of MVPA did not display a significant risk for obesity based on different indicators, even when they had high values of ST. It is known in the literature that interventions to reduce obesity or maintain weight should consider physical activity as one of their main pillars [37]. Physical activity plays an important role in reducing weight in the short term and controlling it in the long term [38].

Over the last few years and decades, society has undergone major technological changes and consequently increased ST [39,40]. These changes have affected everyone’s lifestyle and caused a large increase in sitting time in the entire population, resulting in a daily decrease of more than 100 calories and accounting for the significant increase in the weight of adults [39]. As a result, people who meet the physical activity recommendations or accumulate large amounts of physical activity during the week may still spend a considerable amount of time in sedentary pursuits [41]. In fact, it is increasingly clear in the literature that physical activity and ST should be treated as distinct variables, even though the joint association of these two behaviors on various health outcomes throughout life must, nevertheless, be analyzed [19,41]. According to Lee et al., ST or physical inactivity is an important health risk factor, which is becoming a public policy concern like obesity and smoking [41]. Therefore, our study sought to analyze whether ST or some amount of it could increase the risk of obesity, even in subjects who reached the MVPA recommendations. The findings show that regardless of the amount of sedentary time, those who met the MVPA recommendations did not have a significantly higher risk for obesity in any of the indicators considered in this study. These results show that reaching 300 min/week of MVPA is an important step toward obesity prevention. On the other hand, subjects who did not reach 300 min/week had a significantly higher risk for obesity based on at least one of the obesity indicators. This risk was observed across the ST quartiles, demonstrating that the amount of ST was not a stronger predictor of obesity than the compliance with the MVPA recommendations.

The results did not show a proportional increase in the risk of obesity as MVPA decreased or across different categories of ST between groups. This reflects the difficulty of obtaining a cut-off value for ST that protects against obesity. Our findings do not allow us to state that ST increases the risk of obesity. To date, there is no consensus on the exact upper time recommendation for sedentary behavior. In this regard, the WHO only recommends limiting the amount of time spent in sedentary behavior [4]. To verify whether the risk increased with the increase in sedentary time, we adopted the use of quartiles to define the categories of sedentary time. The publication by Stamatakis et al. 2019, which served as a guideline for the design of the present study, adopted four categories of sitting time (<4 h/d; 4- < 6 h/d; 6- < 8 h/d; and ≥8) [30]. Stamatakis et al. 2019, however, used the International Physical Activity Questionnaire (IPAQ) to determine ST, while the present study measured ST via accelerometers [30]. Questionnaires can underestimate ST, and as we used accelerometers, we chose to adopt different cut-off values, aiming at a more egalitarian and adequate distribution of individuals by category, after having seen that the distribution was not in agreement [42,43]. In addition, the term ST was adopted instead of sitting time, due to the difference in measurement instruments [42,43]. Another article considered for the design of the present study was the one by van der Ploeg et al., which determined the relationship between sitting time and all-cause mortality, analyzing the exposure of sitting time in groups with different levels of physical activity [31]. The authors also used data from questionnaires to obtain values for sitting time and physical activity. Sitting time was divided into 0- < 4, 4- < 8, 8- < 11, and 11 or more hours a day, which differs from our categories those of Stamatakis et al., 2019 [31].

The study by Stamatakis et al. investigated the risk of different categories of sitting time and MVPA in all-cause and cardiovascular disease deaths [30]. The findings showed that for the groups that met the physical activity recommendations, the risk of sitting time became insignificant. However in the less active groups (<150 MVPA min/week), there appeared to be almost a dose–response association [30]. These findings corroborate our findings, where no increased risk was found for those who met the MVPA recommendations. According to the results obtained, we understand that the level of physical activity was more important than ST regarding the risk of obesity. One of the mechanisms linked to the association between increased intensities of physical activity and reduced body adiposity is that increased intensities of physical activity may also contribute to a reduction in the craving of high-fat foods [44].

It was not possible to obtain any dose–response or linear relationship between ST and the risk of obesity, as the results differed across the various indicators of obesity and did not follow a proportional dose–response relationship. Following the same line of research, van der Ploeg et al. showed that sitting for a long time was significantly associated with a higher risk of all-cause mortality, regardless of physical activity. The authors, however, considered a different outcome than ours [31]. It should be noted that in this study, it can be argued that reaching the physical activity recommendations almost entirely nullified the risks caused by sitting time. An important meta-analysis of physical activity measured by accelerometers and ST with more than 44,000 adults and elderly people showed that the risk of death appeared to be reduced, although not completely eliminated, for people in the highest MVPA groups. At the same time, lower MVPA groups presented greater risks, and these risks became greater as ST increased [43].

Although we did not find any dose–response relationship between ST and obesity in the present study, we understand that, as suggested in the term “physiology of inactivity”, sitting too much is not the same as lack of exercise. Therefore, more studies are essential to investigate the effects and consequences of the amount of ST on health in groups with different levels of physical activity [19,43,45]. The present study showed that the level of physical activity was more incisive in the risk of obesity, but the deleterious effects of ST on health should not be disregarded. We reiterate that the findings of this study further reinforce the importance of meeting the MVPA recommendations rather than dismissing the possible harmful effects of excessive ST. The amount of physical activity at different intensities (light, moderate, vigorous, and total) analyzed together with ST may also present different results for the risk of obesity, and other associations should be investigated in future studies, as previously suggested [43,46]. According to Matthews et al., replacing at least 1 h of ST with light intensity physical activity is enough to reduce the risk of mortality by 18% [46]. This substitution is particularly valuable for those who do not meet the MVPA recommendations.

This study has some strengths, including the use of accelerometers in developing countries to measure the level of physical activity, a method considered the gold standard for obtaining data on the amount and intensity of physical activity in epidemiological studies. The sample size, containing data from eight Latin American countries and the utilization of standardized methods, should also be valued. The limitations of the study, however, need to be considered as well. The study only contains data from urban areas of the participating countries and therefore cannot be considered representative at the national level. Furthermore, the cross-sectional design precludes determining causality.

Public policies should reinforce the need to achieve the MVPA recommendations and more action plans that favor the achievement of the MVPA recommendations, thus “vaccinating“ the population against the harm that can be caused by ST in a world that favors this behavior in the face of technological advances.

## 5. Conclusions

In this study, we concluded that achieving the MVPA recommendations protects against obesity, regardless of ST. Subjects who did not meet the MVPA recommendations tended to have a higher risk of obesity, based on waist and neck circumferences. This indicates that MVPA has a stronger association with obesity than ST. For people who undergo large amounts of ST, meeting the MVPA recommendations proves to be a good antidote. Future studies should investigate the necessary amount of physical activity at other intensities or the number of walking steps that may protect against obesity, independently of ST. In addition, further studies are needed to understand whether there is an amount of ST that can override the benefits of physical activity. It would also be of great value to investigate the dosage of MVPA and ST on other health outcomes.

## Figures and Tables

**Figure 1 ijerph-20-05562-f001:**
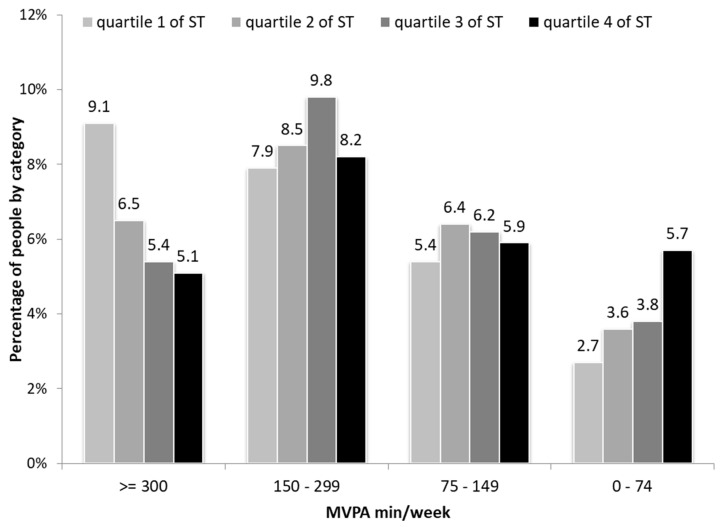
Percentage of sedentary time and moderate-to-vigorous physical activity according to joint categories.

**Figure 2 ijerph-20-05562-f002:**
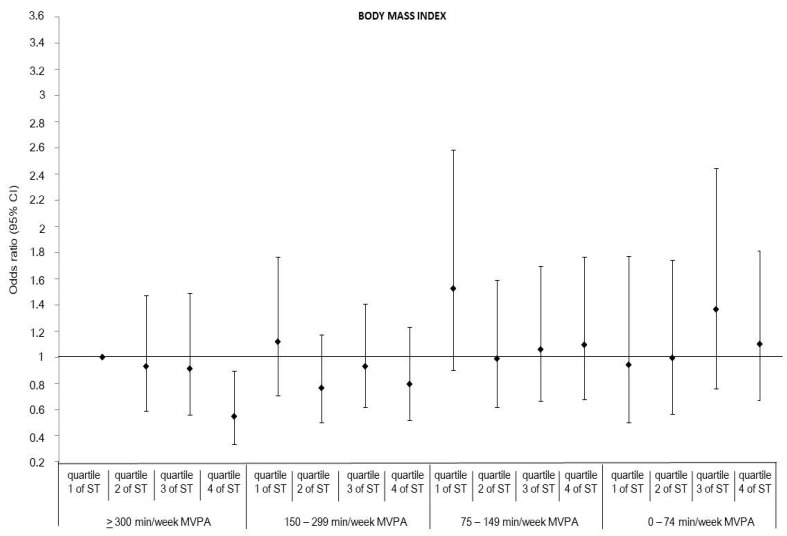
Joint associations of sedentary time and moderate-to-vigorous physical activity with overweight/obesity by body mass index. The analyses were adjusted for country, sex, age, marital status, socioeconomic status, and race/ethnicity.

**Figure 3 ijerph-20-05562-f003:**
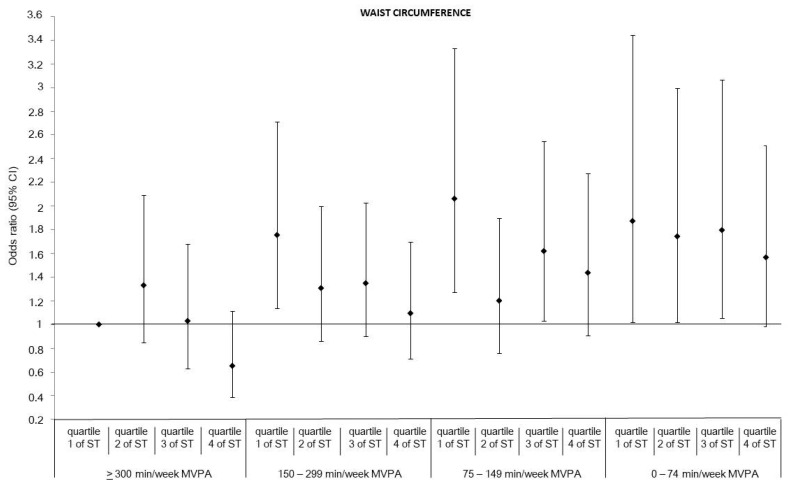
Joint associations of sedentary time and moderate-to-vigorous physical activity with obesity by waist circumference. The analyses were adjusted for country, sex, age, marital status, socioeconomic status, and race/ethnicity.

**Figure 4 ijerph-20-05562-f004:**
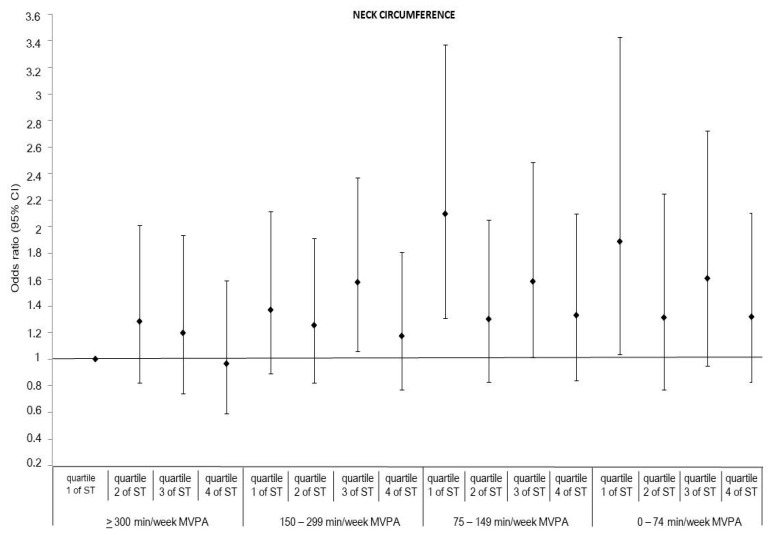
Joint associations of sedentary time and moderate-to-vigorous physical activity with obesity by neck circumference. The analyses were adjusted for country, sex, age, marital status, socioeconomic status, and race/ethnicity.

**Table 1 ijerph-20-05562-t001:** Descriptive analysis (mean (SD) and frequency (%)) of demographic, behavioral, and health status factors.

Variables	Total (N = 2406)	Quartile 4 of ST and 0–74 min/week of MVPA (N = 137)	Quartile 1 of ST and ≥300 min/week of MVPA (N = 218)
Gender (women)	1290 (53.6)	80 (58.4)	87 (39.9)
Age (mean (SD))	39.3 (12.9)	43.4 (13.0)	39.7 (11.7)
Marital Status (n (%))			
Single	849 (35.3)	43 (31.4)	60 (27.5)
Married	1297 (53.9)	67 (48.9)	135 (61.9)
Widowed	78 (3.2)	9 (6.6)	5 (2.3)
Divorced	182 (7.6)	18 (13.1)	18 (8.3)
Country (n (%))			
Argentina	265 (11.0)	21 (15.3)	19 (8.8)
Brazil	501 (20.9)	26 (19.0)	54 (24.8)
Chile	265 (11.0)	4 (2.9)	37 (17.0)
Colombia	305 (12.7)	21 (15.3)	20 (9.1)
Costa Rica	236 (9.8)	10 (7.3)	20 (9.1)
Ecuador	232 (9.6)	11 (8.1)	31 (14.2)
Peru	284 (11.8)	28 (20.4)	22 (10.1)
Venezuela	318 (13.2)	16 (11.7)	15 (6.9)
Socioeconomic status (n (%))			
Low	1219 (50.7)	73 (53.3)	133 (61.0)
Medium	938 (39.0)	41 (29.9)	74 (33.9)
High	249 (10.3)	23 (16.8)	11 (5.1)
Race/ethnicity (n (%))			
Mixed/Caucasian	1158 (51.1)	64 (49.2)	109 (55.6)
Black	143 (6.3)	7 (5.4)	15 (7.7)
White	801 (35.3)	51 (39.2)	53 (27.0)
Others	168 (7.4)	8 (6.2)	19 (9.7)
ST, min/day (mean (SD))	566.5 (120.5)	726.1 (88.2)	410.9 (58.2)
MVPA, min/day (mean (SD))	34.0 (22.9)	6.7 (3.0)	73.7 (28.4)
Body mass index (n (%))			
Overweight/obesity	1575 (65.5)	99 (72.3)	141 (64.6)
Waist circumference (n (%))			
Abdominal obesity	1060 (44.1)	71 (51.8)	83 (38.1)
Neck circumference (n (%))			
Obesity	1029 (42.8)	63 (46.0)	82 (37.6)

MVPA = moderate to vigorous physical activity; ST = sedentary time; min = minutes; and SD = standard deviation.

## Data Availability

The study database is not available for general use due to the terms of consent/assents signed by the participants. It is suggested to contact the author by correspondence for more information or availability of information.

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
