# Peer review of "Joint Association between Sedentary Time and Moderate-to-Vigorous Physical Activity with Obesity Risk in Adults from Latin America"

_ijerph, 2023, doi:10.3390/ijerph20085562_

Round 1
Reviewer 1 Report
A few lines in, makes argument for relevancy of the study. In Latin America most studies used self-report.
In introduction, author's bring up my concern over novelty of the study, mainly that it has already been done, joint associations with ST and MPVA have been done, but this type of research is not prevalent in Latin America.
Covariates are introduced as country,sex, age ,martial status, soecieconomic status and race / ethnicity, but not much is explained as to why these were chosen. I would have liked to see a table showing the relationship between these variables and the outcome variables. Or in a more concise fashion a synopsis saying there were no significant relationships or that the variables were chosen for historic reasons to match previous analyses, etc.
Author Response
We would like to thank for taking the time to review our manuscript and we appreciate their constructive and thoughtful comments. Based on these suggestions, we have revised our manuscript accordingly. Please find below our point-by-point responses to the comments and suggestions. Other changes were put in as per the correction suggested by the other reviewers.
A few lines in, makes argument for relevancy of the study. In Latin America most studies used self-report.
Author’s response: Thanks for the comment. We inserted this statement in the last paragraph of the introduction.
In introduction, author's bring up my concern over novelty of the study, mainly that it has already been done, joint associations with ST and MPVA have been done, but this type of research is not prevalent in Latin America.
Author’s response: Thanks for the comment. We corrected this excerpt as suggested and highlighted that it is a non-common type of research in Latin America and not unprecedented as previously stated. This correction is in the last paragraph of the introduction (together with the correction mentioned in the previous comment).
Covariates are introduced as country,sex, age ,martial status, soecieconomic status and race / ethnicity, but not much is explained as to why these were chosen. I would have liked to see a table showing the relationship between these variables and the outcome variables. Or in a more concise fashion a synopsis saying there were no significant relationships or that the variables were chosen for historic reasons to match previous analyses, etc.
Author’s response: Thanks for your comment. We selected the covariates based on previously published studies (bellow). As there are 16 categories of physical activity and sedentary time, the authors understand that performing a comparison analysis of categories with sociodemographic variables using the chi-square would leave the analysis very weak due to the number of physical activity and sedentary time categories. In addition, it would have the other categories of sociodemographic variables. For this reason, the authors chose to present only a descriptive analysis in Table 1. Anyway, descriptive analyzes of the physical activity and sedentary time with covariates were published in other articles.
https://pubmed.ncbi.nlm.nih.gov/32756330/
https://pubmed.ncbi.nlm.nih.gov/31603392/
https://pubmed.ncbi.nlm.nih.gov/32343753/
https://pubmed.ncbi.nlm.nih.gov/33502675/
Reviewer 2 Report
1. The manuscript reports the association between sedentary time and moderate-to-vigorous physical activity with obesity risk in adults from Latin America.
2. While the authors appear to apply a thorough methodology, the results are rather general and vague and not providing many contributions. This might be due to the research questions being very broad but also phrased in a way in which it cannot be answered through the methodology.
3. There is a keen argument that the first ten reported countries in the world with the highest number of obese citizens are due to their high population densities. Won’t it be ideal to compare the data with smaller countries and compare the proportions of people that are obese? The 60% reported by authors are for the continent rather than for each of the countries in Latin America.
4. On sampling, what is the rationale for choosing those eight countries? How could the authors prove that their samples are representative of the Latin American population? What is the sample size for this study etc?
5. Why did the authors use age and socioeconomic factors as predictors? Why not gender, location etc?
6. Line 114 -119 – The authors report that there are different cut-offs for obesity of young adults (15-19), why were they included in the first place, rather than screening them out at the later stage?
7. In addition, the manuscript needs to demonstrate more evidence of a detailed understanding of research techniques consistent with advanced academic enquiry in the nutrition discipline, or creative domain to which the research relates. This was an area of concern when reading and reviewing the manuscript. Better coherence and logical argument are needed. For example, it is not enough to say no studies have reported the relationship between MVPA and ST and obesity. Why then your study? Why did it worth doing?
8. There must be a clear explanation on how the participants were recruited, perhaps at the beginning of section 2, too late to state this in Line 185. I suggest you add a subsection as ‘Data collection procedure’.
9. On data analysis, why was Chi square not used to determine the level of relationship between the variables? Due to the nature of your data, I also suggest that you find the False Discovery Rate odd ratio that would be significant at a p-value omnibus. This is a 3-level regression analysis which would help you in your 2 model predictors (MVPA and ST).
10. Line 204, this was so because you used omnibus p>.5
11. Table 1, why was information on participants’ marital status collected? Was there any difference between how obesity was measured using BMI and Neck circumference?
12. Lines 315 -322 were irrelevant under the discussion section.
13. I am not surprised to read that the conclusion choices MVPA over ST relative to obesity, common sense could predict that.
14. Make the contribution of your findings clearer: Write out very clearly and point by point what you conclude, add an implication section - rephrase the abstract so it contains your main conclusion and implication
15. All in all, I miss the breakthrough of this study. It confirms what is known in the field. Subjective, thanks.
*** Joint and Association in your topic are tautology.
Author Response
We would like to thank for taking the time to review our manuscript and we appreciate their constructive and thoughtful comments. Based on these suggestions, we have revised our manuscript accordingly. Please find below our point-by-point responses to the comments and suggestions. Other changes were put in as per the correction suggested by the other reviewers.
The manuscript reports the association between sedentary time and moderate-to-vigorous physical activity with obesity risk in adults from Latin America.
Author’s response: Thanks for the comment.
While the authors appear to apply a thorough methodology, the results are rather general and vague and not providing many contributions. This might be due to the research questions being very broad but also phrased in a way in which it cannot be answered through the methodology.
Author’s response: Thank you for your feedback. We believe that the potential contributions of the findings herein, that is providing an indication of the joint association between sedentary time and physical activity in Latin American adults for the first time, far outweigh any potential limitations. Nonetheless, we acknowledge your comment and have mentioned this as a potential limitation of the study. On the other hand, the strengths of this study include the large sample size, comparable data collection protocols, and the use of objective methods to assess PA and sedentary time – use of these are rare in Latin America countries where most previous research has relied on self-reported instruments. These types of objective assessments for physical activity and sedentary time are rare for population health surveys and the best available evidence must be used to support and guide action to increase physical activity levels.
There is a keen argument that the first ten reported countries in the world with the highest number of obese citizens are due to their high population densities. Won’t it be ideal to compare the data with smaller countries and compare the proportions of people that are obese? The 60% reported by authors are for the continent rather than for each of the countries in Latin America.
Author’s response: Thanks for the comment. Although we understand the importance of studies that investigate geographic, geopolitical and other characteristics related to the environment and obesity, the present study did not aim to compare obesity between countries. Our study aimed to analyze the joint association of sedentary time and moderate-vigorous physical activity with different indicators of obesity. This comment of yours gives us light for a future study that analyzes and compares obesity according to geographic and population characteristics, which would be of great importance for public policies in Latin America. I share here some studies of our research group with this theme that have already been published.
https://pubmed.ncbi.nlm.nih.gov/34371915/
https://pubmed.ncbi.nlm.nih.gov/32987637/
https://pubmed.ncbi.nlm.nih.gov/35162152/
On sampling, what is the rationale for choosing those eight countries? How could the authors prove that their samples are representative of the Latin American population? What is the sample size for this study etc?
Author’s response: Thanks for the comment. We inserted more details about the project in the article, in addition to adding other information commented during the review about the methods.
https://www.ncbi.nlm.nih.gov/pmc/articles/PMC4736497/
Why did the authors use age and socioeconomic factors as predictors? Why not gender, location etc?
Author’s response: We selected the covariates based on previously published studies (bellow). The analyses were adjusted for country, sex, age, marital status, socioeconomic status and race/ethnicity.
https://pubmed.ncbi.nlm.nih.gov/32756330/
https://pubmed.ncbi.nlm.nih.gov/31603392/
https://pubmed.ncbi.nlm.nih.gov/32343753/
https://pubmed.ncbi.nlm.nih.gov/33502675/
Line 114 -119 – The authors report that there are different cut-offs for obesity of young adults (15-19), why were they included in the first place, rather than screening them out at the later stage?
Author’s response: Thanks for the question. The authors preferred to exclude adolescents from the sample to work only with a sample of adults. The cutoff of 19 years is justified because after that age, the value considered for obesity classification is the same up to 65 years.
There must be a clear explanation on how the participants were recruited, perhaps at the beginning of section 2, too late to state this in Line 185. I suggest you add a subsection as ‘Data collection procedure’.
Author’s response: Thanks for the comment. We have inserted a subsection on methods as requested and also provided more details on the sampling procedure.
On data analysis, why was Chi square not used to determine the level of relationship between the variables? Due to the nature of your data, I also suggest that you find the False Discovery Rate odd ratio that would be significant at a p-value omnibus. This is a 3-level regression analysis which would help you in your 2 model predictors (MVPA and ST).
Author’s response: Thanks for your comment. As there are 16 categories of physical activity and sedentary time, the authors understand that performing a comparison analysis of categories with sociodemographic variables using the chi-square would leave the analysis very weak due to the number of physical activity and sedentary time categories. For this reason, the authors chose to present only a descriptive analysis in Table 1. About the new regression analysis, we agree with the chief editor as the logistic regression is a statistical analysis method to predict a binary outcome, such as yes or no (obesity or no), based on prior observations of a data set (physical activity and sedentary time). For this reason, we decided to keep the current regression analysis done in the article.
https://pubmed.ncbi.nlm.nih.gov/32756330/
https://pubmed.ncbi.nlm.nih.gov/31603392/
https://pubmed.ncbi.nlm.nih.gov/32343753/
https://pubmed.ncbi.nlm.nih.gov/33502675/
Table 1, why was information on participants’ marital status collected? Was there any difference between how obesity was measured using BMI and Neck circumference?
Author’s response: Thanks for the question. To obtain information on demographic data in the project, a questionnaire was used to collect information on age, gender, years of schooling, number of people in the household, race/ethnicity, marital status and number of years living in the country. This information enriches the project and has been used in several studies already published with the project data.
https://pubmed.ncbi.nlm.nih.gov/32785188/
https://pubmed.ncbi.nlm.nih.gov/32961771/
As for differences in how obesity was measured by BMI and neck circumference, we used classifications from previous studies on the subject. In the case of BMI, we followed the WHO recommendation and participants were classified into underweight/eutrophic (≤24.9 kg/m2) or overweight/obese (≥25.0 kg/m2). In neck circumference, men with a neck circumference >38.5 cm and women with a neck circumference >34.5 cm were classified as obese.
https://pubmed.ncbi.nlm.nih.gov/36647028/
BMI classification: https://pubmed.ncbi.nlm.nih.gov/11234459/
Neck circumference classification: https://pubmed.ncbi.nlm.nih.gov/19010573/
Lines 315 -322 were irrelevant under the discussion section.
Author’s response: Thanks for the comment. We prefer to keep this excerpt from the discussion, as it was these articles cited in the excerpt that inspired the present study. The aforementioned study, as well as ours, analyzed the consequences of different times of sitting with different levels of physical activity. The difference between the studies is that ours is associated with obesity and theirs with all-cause mortality. Furthermore, we commented that the authors categorized sitting time differently than ours. This difference must have two reasons: 1) they used keyboard and accelerometer nodes; 2) due to the tool used to measure, they consider sitting time. Our study prefers the term sedentary time, due to the way accelerometers are categorized (as presented in the study methods).
I am not surprised to read that the conclusion choices MVPA over ST relative to obesity, common sense could predict that.
Author’s response: Thanks for the comment. Sedentary time or sitting time has increasingly drawn the attention of researchers in the area, however investigations on this subject are not yet as thorough as in the case of physical activity and its different intensities. Our hypothesis was that achieving MVPA recommendations could nullify the risk of obesity for those with lower ST, however, for those with excess ST, the risk would only attenuate. We added this at the end of the introduction at the request of another reviewer. The literature has increasingly shown that sedentary time increases the risk of obesity, but it is still unclear whether there is an amount of time that would nullify the benefits of physical activity. In addition, some studies suggest that sedentary time increases the risk of obesity regardless of the level of physical activity. We believe that our study can add to other studies on this gap and serve as a basis for future discussions. Here are some published studies that contributed to the authors' hypothesis.
https://pubmed.ncbi.nlm.nih.gov/23359096/
https://pubmed.ncbi.nlm.nih.gov/30997733/
https://pubmed.ncbi.nlm.nih.gov/27604822/
https://pubmed.ncbi.nlm.nih.gov/31235160/
https://cdnsciencepub.com/doi/full/10.1139/apnm-2020-0272?rfr_dat=cr_pub++0pubmed&url_ver=Z39.88-2003&rfr_id=ori%3Arid%3Acrossref.org
Make the contribution of your findings clearer: Write out very clearly and point by point what you conclude, add an implication section - rephrase the abstract so it contains your main conclusion and implication
Author’s response: Thanks for the suggestion. We reworked the study abstract and improved the main conclusion of the abstract. In addition, we reformulate the conclusion of the article and insert the implications of the findings as suggested.
All in all, I miss the breakthrough of this study. It confirms what is known in the field. Subjective, thanks.
Author’s response: Thanks for the comment. As we mentioned in comment 13, the joint relationship between physical activity and sedentary time with health is still unclear. The literature itself is still controversial on this subject. Even so, the importance of scientific knowledge to confirm or not subjective truths or common sense must be highlighted.
Joint and Association in your topic are tautology.
Author’s response: Thanks for the comment. Several studies have adopted this term for analysis of two variables together for an outcome. Below are some examples of publications that also used this term.
https://pubmed.ncbi.nlm.nih.gov/28544795/
https://www.bmj.com/content/378/bmj-2022-070688
https://pubmed.ncbi.nlm.nih.gov/33580798/
Reviewer 3 Report
The authors must be commended for carrying out a study regarding the association between sedentary time and moderate to vigorous physical activity with obesity risk in adults from Latin America. This topic is significant, the research methodology used in the study is appropriate, the manuscript is written with good clarity, and the study results are very interesting and could be important for improving the health status of the general population. I have only a few suggestions.
Abstract
Please add statistics used in the study.
Line 50 and 51: Please rewrite this sentence. I suggest using a sentence from the conclusion ‘’It is concluded that achieving the MVPA recommendations protects against obesity, regardless of ST.’’ which very clearly describes, in one sentence, the study conclusions.
Introduction
Please add a hypothesis.
Methods
Line 129: Did you track somehow a water-based activity (questionary e.g.)?
Author Response
We would like to thank for taking the time to review our manuscript and we appreciate their constructive and thoughtful comments. Based on these suggestions, we have revised our manuscript accordingly. Please find below our point-by-point responses to the comments and suggestions. Other changes were put in as per the correction suggested by the other reviewers.
The authors must be commended for carrying out a study regarding the association between sedentary time and moderate to vigorous physical activity with obesity risk in adults from Latin America. This topic is significant, the research methodology used in the study is appropriate, the manuscript is written with good clarity, and the study results are very interesting and could be important for improving the health status of the general population. I have only a few suggestions.
Author’s response: Thanks for the comments and compliments. Their suggestions were considered and duly included in the study.
Abstract: Please add statistics used in the study.
Author’s response: Thanks for the suggestion. We have reformulated the abstract and added the analysis used in the study.
Abstract: Line 50 and 51: Please rewrite this sentence. I suggest using a sentence from the conclusion ‘’It is concluded that achieving the MVPA recommendations protects against obesity, regardless of ST.’’ which very clearly describes, in one sentence, the study conclusions.
Author’s response: Thanks for the suggestion. The conclusion sentence of the abstract was duly corrected as suggested.
Introduction: Please add a hypothesis.
Author’s response: Thanks for the suggestion. The authors' hypothesis was inserted at the end of the introduction.
Methods. Line 129: Did you track somehow a water-based activity (questionary e.g.)?
Author’s response: Thanks for the question. Unfortunately, the accelerometer does not have water resistance, making any measurement impossible in water activities. In addition to measurements using accelerometers, the project also measured physical activity using the International Physical Activity Questionnaire (IPAQ) - long version, however, the questionnaire does not provide us with specific information about aquatic activities.
Round 2
Reviewer 2 Report
I have the following suggestions for the authors.
Line 39-40 - Rather than starting the abstract with the aim of the study, the authors could start the abstract with a sentence like
‘Recent studies have shown various relationships between physical activity and the incidence of obesity, but this study critically explored the association of sedentary time (ST) and moderate-to-vigorous physical activity (MVPA) with obesity risk in adults from eight Latin American countries'.
Lines 50-51 should be rewritten as (the new sentence is too authoritative)
This study suggests that achieving the MVPA recommendations will likely protect against obesity, regardless of ST.
Line 90 – scarce should be replaced with ‘limited’ and In addition, be replaced with ‘For example’…
……still limited in Latin America. For example, there are few studies that …..
Line 100 -101 – how can you be too sure that the effects would be nullified? The author could say
…recommendations are very likely to minimise the risks of obesity …….
Line 383 – The first sentence in the conclusions ‘must’ be rewritten as suggested above. ‘It is concluded’ could be replaced with words like ‘In this study, MVPA recommendations were practical in …….
I also suggest that this manuscript be proofread by someone with a good knowledge of the English language, as there are still very many syntaxes (grammar) errors in the manuscript
Author Response
We would like to thank for taking the time to review our manuscript and we appreciate their constructive and thoughtful comments. Based on these suggestions, we have revised our manuscript accordingly. Please find below our point-by-point responses to the comments and suggestions. Other changes were put in as per the correction suggested by the other reviewers.
Coments:
I have the following suggestions for the authors.
Line 39-40 - Rather than starting the abstract with the aim of the study, the authors could start the abstract with a sentence like:
‘Recent studies have shown various relationships between physical activity and the incidence of obesity, but this study critically explored the association of sedentary time (ST) and moderate-to-vigorous physical activity (MVPA) with obesity risk in adults from eight Latin American countries'.
Author’s response: Thanks for the sugestion. The correction was made according to the request.
Lines 50-51 should be rewritten as (the new sentence is too authoritative)
This study suggests that achieving the MVPA recommendations will likely protect against obesity, regardless of ST.
Author’s response: Thanks for the sugestion. The sentence was rewritten as suggested.
Line 90 – scarce should be replaced with ‘limited’ and In addition, be replaced with ‘For example’…
……still limited in Latin America. For example, there are few studies that …..
Author’s response: Thanks for the sugestion. Terms have been changed as suggested.
Line 100 -101 – how can you be too sure that the effects would be nullified? The author could say
…recommendations are very likely to minimise the risks of obesity …….
Author’s response: Thanks for the sugestion. We corrected the sentence as suggested.
Line 383 – The first sentence in the conclusions ‘must’ be rewritten as suggested above. ‘It is concluded’ could be replaced with words like ‘In this study, MVPA recommendations were practical in …….
Author’s response: Thanks for the correction. This phrase was suggested by the other reviewers. We try to accommodate requests from all reviewers.
I also suggest that this manuscript be proofread by someone with a good knowledge of the English language, as there are still very many syntaxes (grammar) errors in the manuscript
Author’s response: Thanks for the sugestion. We respond to your request and send the article to a translator specialized in the language.
